

# Assessing the opportunity for selection to impact morphological traits in crosses between two *Solanum* species

Jorja Burch[1], Crystal Nava[1] and Heath Blackmon[1,2]

[1] Biology, Texas A&M University, College Station, Texas, United States
[2] Interdisciplinary Program in Ecology and Evolutionary Biology, Texas A&M University, College Station, Texas, United States

## ABSTRACT

Within biology, there have been long-standing goals to understand how traits impact fitness, determine the degree of adaptation, and predict responses to selection. One key step in answering these questions is to study the mode of gene action or genetic architecture of traits. The genetic architecture underlying a trait will ultimately determine whether selection can lead to a change in the phenotype. Theoretical and empirical research have shown that additive architectures are most responsive to selection. The genus *Solanum* offers a unique system to quantify the genetic architecture of traits. Crosses between *Solanum pennellii* and *S. lycopersicum*, which have evolved unique adaptive traits for very different environments, offer an opportunity to investigate the genetic architecture of a variety of morphological traits that often are not variable within species. We generated cohorts between strains of these two *Solanum* species and collected phenotypic data for eight morphological traits. The genetic architectures underlying these traits were estimated using an information-theoretic approach to line cross analysis. By estimating the genetic architectures of these traits, we were able to show a key role for maternal and epistatic effects and infer the accessibility of these traits to selection.

## INTRODUCTION

Much of plant breeding is focused on selecting for traits that improve crop yield and survival in adverse environments (*Fischer, 2001*). The success of selection for specific traits is dependent on the genetic architecture underlying a trait, specifically the proportion of additive genetic effects that are present within the genetic architecture. It has been shown that traits more associated with fitness, those often the focus of selective breeding, have genetic architectures that contain a significant proportion of epistatic genetic effects (*Burch et al., 2024*). This excess of epistatic genetic effects is hypothesized to result from the exhaustion of additive genetic effects, as beneficial alleles are fixed (*Burch et al., 2024*; *Fisher, 1930*; but see: *Laurie et al., 2004*; *Walsh & Lynch, 2018*). By quantifying a trait's genetic architecture, we can understand how traits will respond to selection.

Genetic architecture is a term that is commonly misunderstood and has a variety of interpretations. For this article, genetic architecture refers to the mode of gene action (*e.g.*,

Corresponding author
Heath Blackmon,
blackmon@tamu.edu

additive, dominance, epistatic, *etc.*). In other words, genetic architecture describes how variation in genotypes map to variation in phenotypes (*Blackmon & Demuth, 2016*). Quantifying the genetic architecture of a trait that has diverged between two species can be challenging, especially when identifying the contribution of epistatic genetic effects. Quantitative Trait Locus (QTL) and Genome-Wide Association Studies (GWAS) studies are effective in determining regions of the genome responsible for trait variation (*Frary et al., 2010*; *Liu, Jiang & Li, 2023*), but these methods are less effective at detecting regions containing epistatic components of the genetic architecture underlying a trait (*Demuth & Wade, 2006*; *Laurie et al., 2014*; *Mackay, 2001*). For this reason, alternative approaches that dispense with the attempt to localize the genes that impact a trait but instead focus on the modes of gene action that impact traits are useful to understand how traits may respond to selection.

A commonly used method of quantifying genetic architecture is line cross analysis (LCA). LCA is a quantitative genetics approach where crosses are made between two parental strains to create an F1. The F1 is then crossed with the parents to create a series of backcrosses, and the mean and standard error of phenotypes for each cohort is used to estimate the composite genetic effects underlying traits' divergence. This method can be applied in any system where fertile offspring can be made between the original parents (*e. g.*, closely related species, strains, lines). Previously, this method relied on a joint-scaling approach, where a simple additive model is fit first, and then more complex genetic effects are applied until there is no significant improvement to the model (*Mather & Jinks, 1971*). However, the order in which genetic effects are applied can impact which effects appear significant. The joint-scaling method also ignores the degree of model selection uncertainty implied by the data. To overcome these limitations, we employ an information-theoretic approach (*Armstrong, Anderson & Blackmon, 2019*; *Blackmon & Demuth, 2016*). This method uses model averaging to estimate the contribution of the genetic effects to the cohort means while accounting for model selection uncertainty. By understanding the genetic architecture of traits and employing an appropriate method that accounts for model selection uncertainty, we can make inferences about how accessible a trait is to selection (*Laurie et al., 2004*).

Certain genetic architectures are studied more frequently in some clades than others. Maternal genetic effects have been given much more attention in studies of animals than plants (*McAdam et al., 2002*; *Noguera et al., 2019*; *Willham, 1972*). Maternal effects in plants are often ignored, likely owing to previous suggestions about the negligible importance of maternal effects in plants (but see: *Campbell, Dufresne & Sabatinos, 2020*; *Cockerham, 1963*). Studies that do, however, examine maternal effects in plants, often do not partition them into additive and dominant maternal effects (*Eagles & Hardacre, 1979*; *Singh & Murty, 1980*; *Wolf & Wade, 2016*). The *a priori* exclusion of maternal effects could be masking a significant portion of a trait's genetic architecture, leading to an incomplete understanding of the importance of maternal effects in plants and the expected response of traits to natural or artificial selection. Similarly, epistasis is often ignored when only QTL and GWAS approaches are used. Though standing epistatic variation responds poorly to selection, the inference of epistatic effects on a trait implies that the genomes present

contain pairings of alleles that can impact the trait of interest and can be combined in ways to yield directional changes in the phenotype.

To illustrate the genetic architecture of traits and understand the role of maternal genetic effects, we have generated crosses between two species of tomato, *Solanum pennellii* and *S. lycopersicum. Solanum pennellii* is a wild tomato species endemic to Andean regions in South America and has evolved traits that allow it to thrive in dry, nutrient-deficient environments (*Bolger et al., 2014*). The domesticated tomato species, *S. lycopersicum*, is the second most important vegetable crop in the world (*Wakil, Brust & Perring, 2017*). *Solanum pennellii* and *S. lycopersicum* diverged 2.7 million years ago and remain reproductively compatible (but see: *Moyle & Nakazato, 2008*), making them excellent candidates for identifying the genetic architecture underlying trait divergence between the parental cohorts. We have analyzed the genetic architecture of eight morphological traits within crosses between *S. pennellii* and *S. lycopersicum*. These traits include leaf area, leaf perimeter, leaf shape, leaf length, leaf width, leaf roundness, left-to-right areal ratio of the leaf, and seed mass. We quantified the genetic architecture for each trait, where we detected significant epistatic and maternal effects, allowing us to make inferences about how accessible a trait is to selection within crosses of two *Solanum* species.

## MATERIALS AND METHODS

### *Solanum* cohort generation

Seeds for the original cohorts (*S. lycopersicum* (strain VF36), *S. pennellii* (strain 716), and F1 hybrids (strain 4135)) were obtained from the C.M. Rick Tomato Genetics Resource Center at the University of California, Davis. In September 2021, using seeds from the three strains from the resource center, we generated the crosses shown in Table 1. F1 seeds were used to generate the crosses but were unavailable to grow for data collection.

Seeds were treated with a 20% commercial bleach solution for 20 min and then placed in soil for germination. Seedlings were grown in a growth chamber (23 °C, 12-h day/night lighting, and 50% humidity) in four-inch pots. One month after germination, seedlings were transplanted into eight-inch pots and relocated to the experimental greenhouse at Texas A&M University (30.615 N and 96.339 W). The greenhouse temperature was maintained at 23 °C, while lighting and humidity were allowed to fluctuate. After seeds were generated for all cohorts, they were washed, allowed to dry completely, and stored until planted for data collection in August 2022.

### Trait data collection

Seeds were planted in August of 2022. F1 individuals were used for generating the new cohorts, but F1 individuals were not available for data collection after the cohort set had been generated. Three tables in the greenhouse were used and cohorts of each type were distributed equally on all tables, ensuring that no cohorts experienced biased microclimates within the growing area. For the five cohorts being measured (P1, P2, F2, BC1, rBC2), mature branches and leaves were collected 77 days after planting. Specimens were pressed using a plant press for 27 days in a well-ventilated area and then fixed on

**Table 1 Description of crosses.** Crosses were made between the parental cohorts and are denoted in sire-by-dam format. The F1 cohort is listed here to document its genetic makeup, but phenotype measurements were not collected for the F1 in this study as F1 individuals were not available for data collection.

| Cohort | Cross (sire × dam) |
| --- | --- |
| P1 | *S. pennellii*–LA0716 desert parent (self-pollinated) |
| P2 | *S. lycopersicum*–VF36 domestic parent (self-pollinated) |
| F1 | *S. pennellii* × *S. lycopersicum* (LA4135) |
| F2 | F1 (self-pollinated) |
| BC1 | *S. pennellii* × F1 |
| rBC2 | F1 × *S. lycopersicum* |

blotting paper with pH-neutral PVA glue. The specimens were laid on white paper containing a black reference square (1 cm × 1 cm). The specimens were scanned as JPEG images using a Brother DCP L2540DW printer, producing an image like Fig. 1A (left). Scanned images were used for leaf area, leaf perimeter, leaf length, and leaf width measurements. The images were edited using ImageJ (*Schneider, Rasband & Eliceiri, 2012*). The images were cropped, and the leaves were isolated from their stems and other leaves using the "Paintbrush Tool". Despeckling was applied to fill in the leaves to allow for accurate leaf measurements. The threshold was manually adjusted to convert the image to a silhouette, resulting in an image seen in Fig. 1A (right). The leaf midvein was not easily identified in the scanned images, so the samples were photographed using a Canon EOS 2000D camera to obtain images suitable for areal ratio measures (see "Areal Ratio" section for details). These methods were done for 85 leaves for which measurements were taken. With leaves collected from two plants per cohort, there were 23 leaves for P1, 14 leaves for P2, 13 leaves for F2, 14 leaves for BC1, and 21 leaves for rBC2. Given that multiple leaves were taken from each plant, in our statistical analyses the leaves from each plant were treated as non-independent individuals.

*Leaf area:* Leaf area was determined through ImageJ analysis and unit conversions. ImageJ measurements were set to calculate the area and display the results. The leaf was selected from the image with the "Wand (tracing) tool", and the leaf area was found in pixels using the "Measure" command. The area of the reference square was calculated using the same method. The areas were converted to centimeters squared by dividing the pixel count for the leaf area by the pixel count for the 1 cm$^2$ reference square.

*Leaf perimeter:* For leaf perimeter, ImageJ measurements were set to calculate the perimeter and display the results. The leaf was selected from the image with the "Wand (tracing) tool", and the leaf perimeter was found in pixels using the "Measure" command. The perimeter of the reference square was calculated in the same manner. The perimeters were converted to centimeters using the known perimeter of the reference square.

*Leaf perimeter-area ratio:* The ratio of perimeter to area was calculated using the previously determined calculations of leaf perimeter and leaf area as proposed by *Yu et al. (2019)*. The leaf perimeter was divided by leaf area, resulting in units of cm$^{-1}$. We used the

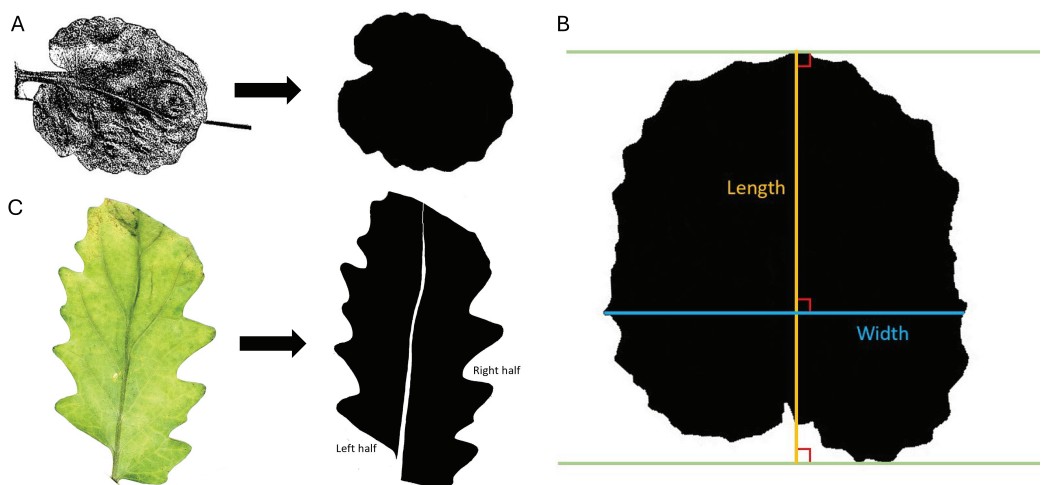

**Figure 1  Trait measurements.** (A) Transition from original scanned leaf image to the edited silhouette, used to determine leaf area, leaf perimeter, leaf width, and leaf length. (B) Example of measurement parameters for leaf length axis (orange) and leaf width axis (blue). Green lines are parallel and touch the farthest points between the protruding base and apex of the leaf. (C) Transition from the original color image of the leaf to the edited silhouette with midvein removed and left and right halves labeled.

leaf perimeter-area ratio as a proxy for quantifying leaf shape. Larger perimeter-area ratios indicate a leaf with more or larger lobes, while smaller perimeter-area ratios indicate a leaf with less or smaller lobes.

*Leaf length:* In ImageJ, leaf length was measured as the longest perpendicular extension from a straight line at the protruding base of the leaf to a parallel line at the apex of the leaf. Leaf length was defined with parameters similar to *Schrader et al. (2021)* and shown in Fig. 1B. ImageJ measurements were set to calculate length and display the results. A line was drawn on the leaf using the "Straight Line tool" and "Angle tool" according to the specifications outlined above, and leaf length was calculated in pixels with the "Measure" command. The length of the reference square (in pixels) was also calculated using the "Straight Line tool" and "Measure" commands. The leaf length was converted to centimeters using the known length of the reference square.

*Leaf width:* In a similar manner to leaf length, leaf width was measured in ImageJ as the longest extension from any two points on the edges of the leaf perpendicular to the length axis (described in the previous section). The leaf width axis was measured according to parameters like *Schrader et al. (2021)* and shown in Fig. 1B. ImageJ measurements were set to calculate width and display the results. A line was drawn on the leaf using the "Straight Line tool" and "Angle tool" according to the specifications outlined above, and leaf width was calculated in pixels using the "Measure" command. The width of the reference square is the same as the length of the reference square found above. The leaf width was converted to centimeters using the known width of the reference square.

*Leaf width-length ratio:* Once the length and width of the leaf were determined, leaf width was divided by leaf length to calculate the ratio of width to length to indicate leaf roundness. The leaf width-length ratio was measured as a proxy for leaf roundness. Ratios

close to zero indicate leaves elongated along the length axis, while ratios larger than one indicate leaves elongated along the width axis, and intermediate ratios close to one indicate more circular leaves.

*Areal ratio:* As a measure of leaf symmetry, the areal ratio of the leaf was determined by finding the ratio of the left half area to the right half area of the leaf, as done by *Yu et al. (2019)*. Each leaf was laid next to a reference square (1 cm × 1 cm) on a flat white surface and photographed using a Canon EOS 2000D camera. The camera was mounted directly above the leaves at a consistent height (approximately 22.5 cm above the laid-out leaves). Direct lighting was applied to the leaves for photographing using an overhead light positioned 35.5 cm above the stage. Auto aperture settings were used for all photos. Canon EOS Utility software was used for remote shooting to minimize vibration. The highest contrast, sharpness, saturation, and color tone were used, resulting in images like Fig. 1C (left).

Using ImageJ, the images were edited with the "Paintbrush Tool" to remove the midvein and split the leaf into the left and right halves. The left and right halves were determined by orienting the image with the apex at the top of the screen and the base at the bottom. The leaves and square were converted to black and white by adjusting the threshold, and they were filled in, where necessary, to be solid black, as seen in Fig. 1C (right). Measurements were set to calculate the area. Each leaf half was selected from the image using the "Wand (tracing) tool", and the area was measured in units of pixels with the "Measure" command. The area of the reference square was calculated using the same method. Leaf areas were converted to centimeters using the known area of the reference square. The area of the left half was divided by the area of the right half to determine the areal ratio of the left half to the right half as an indicator of symmetry.

*Seed mass:* Various seeds from several dates (09/13/2021–11/17/2021) during one seasonal harvest were collected, washed, and allowed to dry completely in a ventilated area on blotting paper. Seeds were individually weighed in grams. Fifteen seeds were weighed for each cohort, with the exception of rBC2, which had 13 seeds.

**Line cross analysis**

The cohorts included in this study define which genetic effects can be inferred. We include both parental cohorts (P1 and P2), an F2 cohort, and two backcrosses (BC1 and rBC2), allowing us to make inferences for ten different genetic effects. Effects include three types of additive genetic effects (autosomal additive, cytotype additive, and maternal effect additive), two types of dominance genetic effects (autosomal dominance and maternal effect dominance), and five types of epistatic interactions (autosomal additive-by-additive, autosomal dominance-by-dominance, autosomal additive-by-dominance, autosomal additive-by-cytotype additive, and autosomal dominance-by-cytotype additive) (Table 2). We recognize this is a relatively small cohort set, however previous studies have shown the ability of LCA to infer epistatic genetic effects with small cohort sets that align with inferences of epistasis within larger cohort sets. *Burch et al. (2024)* analyzed 40 datasets with a cohort set of 16, then reanalyzed the same datasets after they were reduced to a

**Table 2 C-matrix.** Each row is representative of a different cohort originating from the first two parental cohorts in rows one and two. Each column represents a possible composite genetic effect. Column names indicate the type of composite genetic effect, where capital letters are a portion of the genome (A; autosomal, C; cytotype), and lowercase letters represent types of architecture (a; additive, d; dominance). Mea represents maternal effect additive, and Med represents maternal effect dominance. Epistatic effects are denoted by combining these terms. For instance, AaCa represents an epistatic interaction between additive gene action on the autosome and cytotype additive gene action.

| Cohort (sire × dam) | Aa | Ad | Ca | Mea | Med | AaAa | AaAd | AdAd | AaCa | AdCa |
|---|---|---|---|---|---|---|---|---|---|---|
| P1 (P1 × P1) | 1 | 0 | 1 | 1 | 0 | 1 | 0 | 0 | 1 | 0 |
| P2 (P2 × P2) | −1 | 0 | −1 | −1 | 0 | 1 | 0 | 0 | 1 | 0 |
| F2 ((P1 × P2) × (P1 × P2)) | 0 | 0.5 | −1 | 0 | 1 | 0 | 0 | 0.25 | 0 | −0.5 |
| BC1 (P1 × (P1 × P2)) | 0.5 | 0.5 | −1 | 0 | 1 | 0.25 | 0.25 | 0.25 | −0.5 | −0.5 |
| rBC2 ((P1 × P2) × P2) | −0.5 | 0.5 | −1 | −1 | 0 | 0.25 | −0.25 | 0.25 | 0.5 | −0.5 |

smaller cohort set of five. They found a slight, non-significant bias toward detecting less epistasis in datasets with fewer cohorts.

Line cross analysis was used to estimate the contributions of the composite genetic effects (additive, dominance, epistatic, *etc.*) by analyzing the mean and standard error of a trait for each cohort (P1, P2, F2, BC1, rBC2). Since each leaf was treated as an individual, and multiple leaves were collected and measured from each plant, we used an alternative approach to collecting means and standard errors for each dataset. This ensures that leaves measured from the same plant were treated as non-independent, allowing for more accurate quantification of the genetic architecture. Means were collected by taking the mean of all individual measurements from a single plant, repeating that for each plant, then a cohort mean was calculated as the mean of the plant means. For the standard error, we used the following Eq. (1):

$$SE = \sqrt{\sum_{i=1}^{n} \frac{\sigma_i^2}{s_i}} \tag{1}$$

where $\sigma$ is the standard deviation of the trait from samples of a plant, $n$ is the number of plants, and $s$ is the number of samples from that plant.

We used an information-theoretic LCA approach implemented in the R package SAGA 2.0 (*Armstrong, Anderson & Blackmon, 2019*; *Blackmon & Demuth, 2016*). This software automatically generates an appropriate matrix of coefficients (c-matrix), which describes the opportunity for each of the possible composite genetic effects to impact the trait in each cohort (*Lynch & Walsh, 1998*). After the c-matrix is generated, SAGA uses a weighted least squares regression to fit the measured phenotypes with models of the genetic architecture. From these models, we can evaluate the probability for each genetic effect to impact the phenotype of a cohort, thereby inferring the genetic architecture of the trait. The software uses a small sample size corrected version of the standard AIC (AICc) to account for small sample sizes that are typical in LCA studies. AICc was used to calculate model weights and construct a 95% confidence set of models that are used to generate model-averaged results, accounting for model selection uncertainty (*Burnham & Anderson, 2003*). Phenotypic data

was not included for the F1 cohort, so the c-matrix was customized to exclude the F1. Under the analyzed cohort set, the contribution of ten composite genetic effects were evaluated: additive, dominance, cytotype, maternal effect additive and dominance, and epistatic interactions between additive, dominance, and cytotype additive effects (Table 2).

Results were pooled in various ways to illustrate trends for genetic effects in each trait (*e.g.*, additive, dominance, epistatic, maternal, *etc.*). Using criteria proposed by *Blackmon & Demuth (2016)*, genetic effects were excluded if they did not have a variable importance greater than 0.5 and a standard error that excluded zero. Variable importance is the sum of the model weights in the confidence set of models that include that particular composite genetic effect. To allow for comparison among traits, we also scaled the contributions of genetic effects to sum to one. Briefly, we took the absolute values of the contributions, summed all significant genetic effects, and divided the estimated effects by this sum.

## RESULTS

By employing LCA to quantify the genetic architecture of traits, we were able to infer significant genetic effects for seven of the eight observed traits. Epistatic genetic effects were inferred for five of the eight traits. Maternal effects were inferred for two of the eight traits.

Cohort phenotype means and standard errors for all traits are shown in Fig. 2 and Table S1. If the genetic architecture underlying the trait divergence between the parental cohorts was purely additive, all cohorts would fall on the dotted line connecting the P1 and P2. Deviations from the line are expected to be due to other genetic effects, but environmental effects or sampling effects could also lead to deviations from the additive expectation. For six of the eight traits (area, perimeter, length, width, width-length ratio, and seed mass), the most extreme phenotypes are exhibited by the P1 and P2 cohort (Fig. 2). For perimeter-area ratio, the most extreme phenotypes were in the BC1 and rBC2. For areal ratio, the most extreme phenotypes were in the P2 and F2 cohorts.
The magnitude of the standard errors can impact statistical power to detect composite genetic effects. The large error bars for areal ratio (Fig. 2G) could indicate a sample size too small to accurately quantify the means for this trait and thus our power to infer the genetic architecture for this trait may be reduced.

The results of the composite genetic effects for each phenotype determined by LCA are summarized in Fig. 3. One trait, areal ratio, failed to meet the inclusion criteria as it did not have any genetic effects with a variable importance greater than 0.5 and standard error excluding zero. Therefore, it is excluded from Fig. 3.

The leaf area was split with 43% additive effects and 57% epistatic effects, where the additive effects consisted of cytotype additive, and the epistatic effects consisted of additive-by-additive epistasis. Leaf perimeter consisted of 28% additive effects and 72% epistatic effects, with the additive effects being cytotype additive and the epistatic effects being additive-by-additive and additive-by-dominance epistasis. Leaf perimeter-area ratio had only additive-by-dominance epistatic effects. Leaf length had only autosomal additive effects. Leaf width was split with 37% additive effects and 63% epistatic effects, where the additive effects were additive maternal effects, and the epistatic effects were additive-by-

Peer J

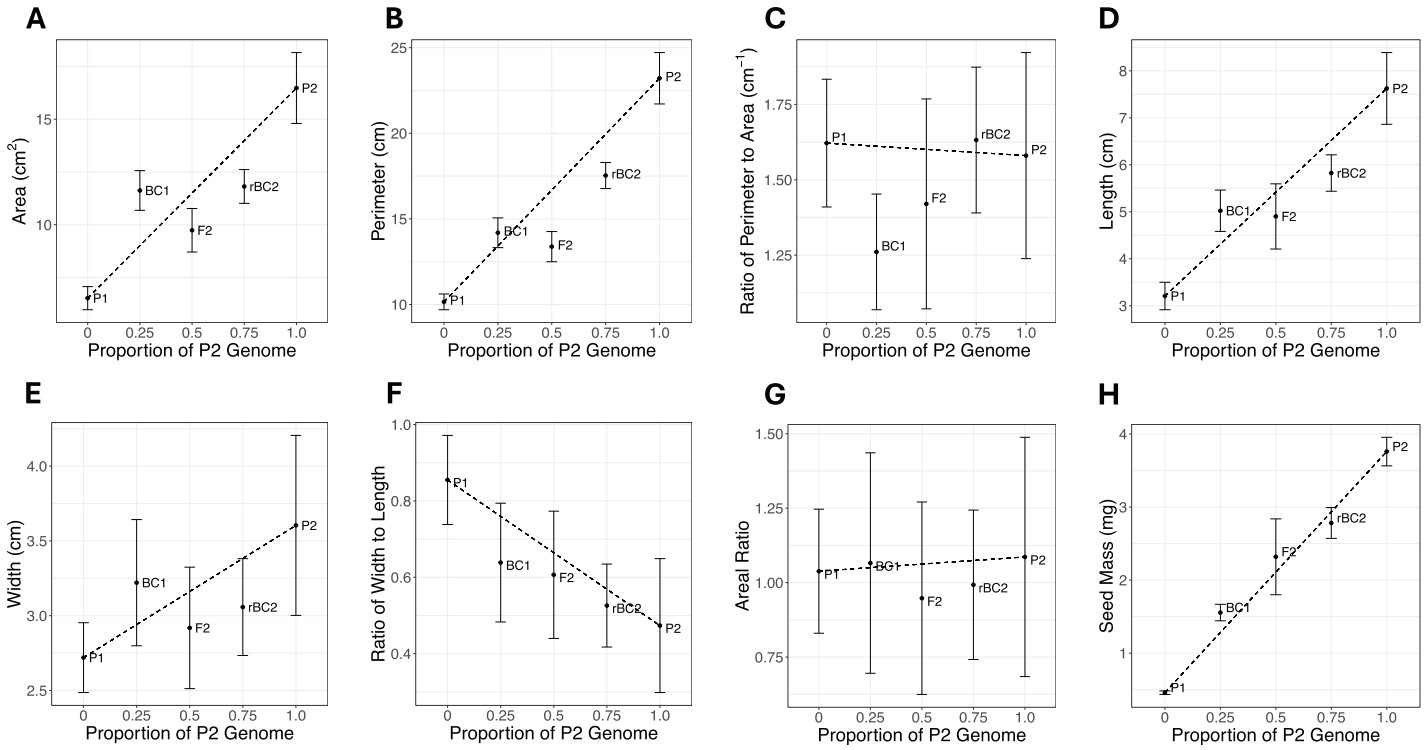

**Figure 2 Mean and standard error plots for each trait.** The horizontal axis represents the proportion of the P2 genome for each cohort, with zero representing 100% P1 genome and one representing 100% P2 genome. The vertical axis represents the phenotype mean. The error bars represent the standard error for each mean. The dotted line between the parental means illustrates the expectation for other cohorts under a purely additive model. (A) Area. (B) Perimeter. (C) Ratio of perimeter to area. (D) Length. (E) Width. (F) Ratio of width to length. (G) Areal ratio. (H) Seed mass.

dominance epistasis. Leaf width-length ratio had only additive effects, which were autosomal additive and cytotype additive. Seed mass was the only phenotype that exhibited dominance effects, but it was still dominated by additive effects (78% additive, 15% dominance, and only 7% epistatic), where the additive effects were autosomal additive, the dominant effects were maternal effect dominance, and the epistatic effects were additive-by-additive epistasis. Areal ratio did not have any genetic effects that were significant, so we were unable to make inferences about the trait's genetic architecture.

With standard errors not overlapping zero and a minimum variable importance of 0.5, all phenotypes displayed significant genetic effects except for areal ratio. Areal ratio did not show significant genetic effects equal to or above the minimum variable importance of 0.5. The maximum variable importance inferred for areal ratio was 0.26 for additive-by-additive epistasis, and the standard error for this composite genetic effect overlapped zero. The proportion of each trait's genetic architecture that displayed epistatic or maternal effects are shown in Table 3.

## DISCUSSION

In this study, we collected plant morphological data, which was used with the quantitative genetics method LCA to quantify the genetic architecture for eight traits. Significant

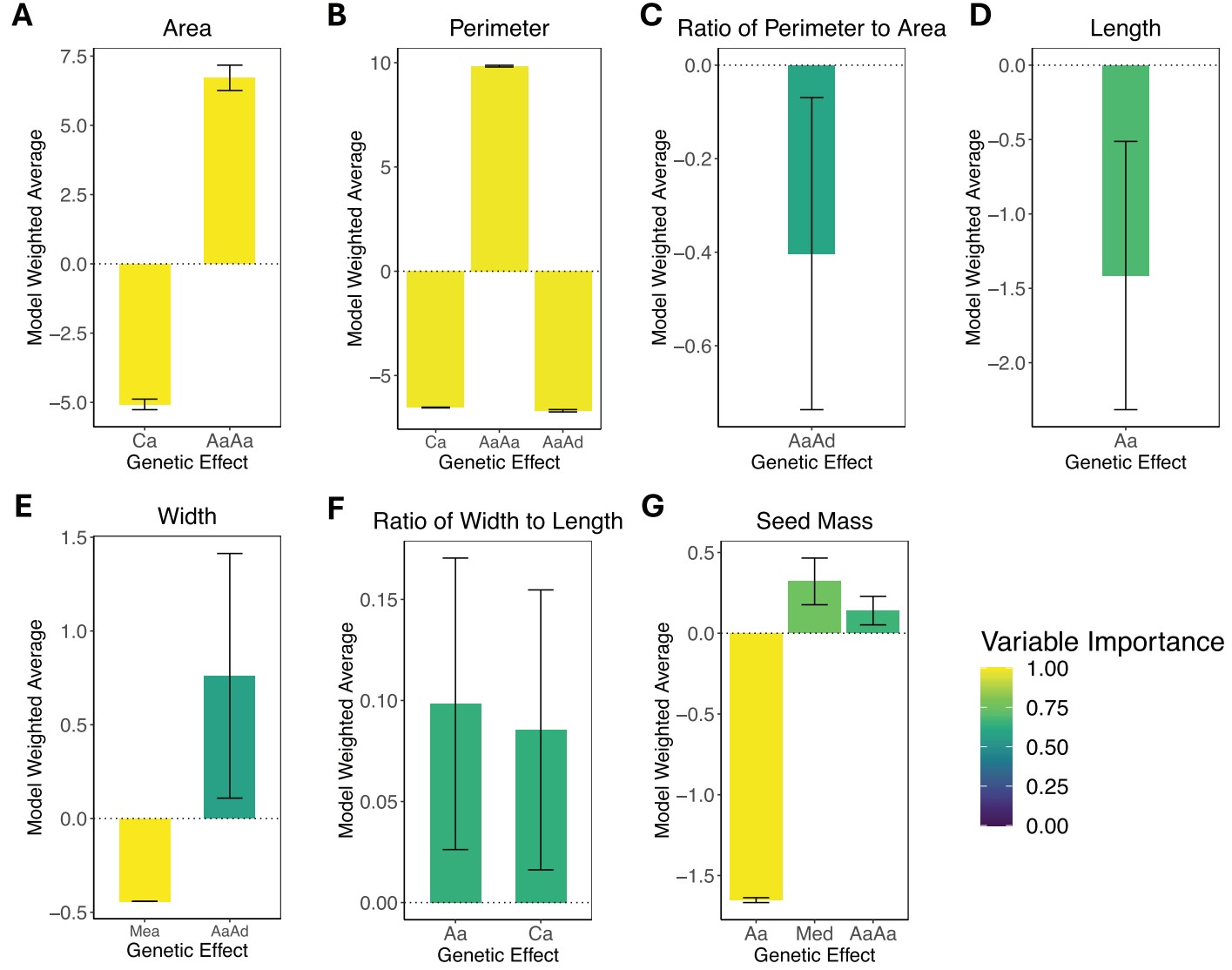

**Figure 3 Composite genetic effects of each trait.** The model weighted average and standard error for each trait's inferred composite genetic effects with variable importance displayed on a color scale. Only the genetic effects that met the inclusion criteria (minimum variable importance of 0.5 and standard error excluding zero) are included in each plot. Note that vertical axes vary among plots due to differences in the magnitude of inferred effects. (A) Area. (B) Perimeter. (C) Perimeter-area ratio. (D) Length. (E) Width. (F) Width-length ratio. (G) Seed mass. The composite genetic effects for areal ratio fall below the minimum variable importance of 0.5 and have standard errors overlapping zero, so they are not included in this figure.

genetic effects were inferred for seven of the eight traits, revealing patterns of additive, epistatic, and maternal effects. By identifying the role of additive genetic effects in each trait's genetic architecture, we can make inferences about how accessible each trait may be to selection.

One advantage of the model averaging approach to LCA is the incorporation of model selection uncertainty. The approach that we used builds a confidence set of models whose AIC weights sum to 0.95. With the cohorts included in our study, 241 possible models of genetic architecture could be constructed. The size of the confidence set of models ranged
**Table 3 Genetic effects for each trait.** Traits are shown with the corresponding proportion of composite genetic effects that were additive (Aa, Ca), dominance (Ad), epistatic (AaAa, AaAd, AaCa, AdAd, AdCa), and maternal (Mea, Med). Both additive and dominance maternal effects are pooled into the total proportion of maternal effects. Only composite genetic effects that met the inclusion criteria were included. NAs indicate the dataset did not have composite genetic effects that met the inclusion criteria. An asterisk (*) indicates that maternal effects (additive and dominance) are excluded for the additive and dominance columns as the total proportion of maternal effects is summarized in the fifth column. The number of models in the confidence model set is shown in the last column.

| Trait | Additive* | Dominance* | Epistatic | Maternal | Conf. model set |
|---|---|---|---|---|---|
| Leaf perimeter-area ratio | 0 | 0 | 1 | 0 | 9 |
| Leaf perimeter | 0.28 | 0 | 0.72 | 0 | 1 |
| Leaf width | 0 | 0 | 0.63 | 0.37 | 7 |
| Leaf area | 0.43 | 0 | 0.57 | 0 | 1 |
| Seed mass | 0.78 | 0 | 0.07 | 0.15 | 3 |
| Leaf length | 1 | 0 | 0 | 0 | 14 |
| Leaf width-length ratio | 1 | 0 | 0 | 0 | 4 |
| Areal ratio | NA | NA | NA | NA | 13 |

from just one for leaf perimeter and leaf area to 14 for leaf length (Table 3). In some cases, this confidence set of models will include models with very different estimates for the same genetic effect, or different genetic effects will be present in each of the models included (*Blackmon & Demuth, 2016*). This occurred for one of our datasets (areal ratio) where no reliable inference of genetic architecture was possible.

For the single dataset where no significant composite genetic effects were inferred, areal ratio, there were 13 models in the confidence set. In our analysis of areal ratio, we found that the top two models (with AIC scores of −16.76 and −16.26) had non-overlapping parameters. The best model included dominant maternal effects and additive-by-dominance epistasis. In contrast, the second-best model (with significant support based on a delta AIC of 0.5) displayed additive-by-additive and additive-by-cytotype epistasis. A possible explanation for the lack of significant composite genetic effects for areal ratio is the large uncertainty in our estimated means (Fig. 2). This may have reduced the ability of LCA to infer significant architectures. Our failure to detect significant composite genetic effects is a clear example of the importance of accounting for model selection uncertainty in LCA. We acknowledge that sample size is a potential limitation to this study, and that increasing the sample size would decrease uncertainty in cohort means and increase power. It is possible that with a larger sample size, we could recover the genetic architecture of areal ratio. Another possibility for a dataset to not produce a significant result is that unmeasured environmental variation impacted the phenotype studied. Line cross analysis assumes that environmental variables are constant across cohorts. However, we think this is unlikely to explain our results as plant positions in our greenhouse were randomized such that all cohorts should have experienced a similar range of environments, and all plants were exposed to the same environment (temperature, humidity, light, *etc.*) within the greenhouse. A final limitation is the number of analyzed cohorts. We included a simple

set of five cohorts (P1, P2, F2, BC1, rBC2). However, we could estimate more genetic effects with a larger set of cohorts, including high-order epistatic interactions that go beyond simple two-partner epistatic effects. We are confident in the genetic effects detected for these traits given the result in Burch et al. (2024), where no significant difference was detected in the amount of epistasis inferred for datasets with a small cohort set (five) vs. a large cohort set (16).

In this study, we have identified an important role for epistasis in the genetic architectures underlying trait divergence. We acknowledge, however, that LCA can infer epistatic genetic effects that have not been experienced by either of the diverging strains or species. For instance, it is possible for two parental strains to only experience additive variation during their divergence, but when we cross them, new multi-locus genotypes that may never have been present during the divergence of the strains are generated. These multi-locus genotypes may produce epistatic genetic effects that LCA recovers. This possibility is greater when the parental cohorts are more diverged (Burch et al., 2024). However, we would argue that the appropriate way to think about LCA experiments is to consider them as an exploration of the genetic effects in a pangenome that includes both the unique and shared alleles of both parental strains. In cases where epistasis is inferred, it shows us that within this novel pangenome, in our case, a pangenome that brings together the genomes of two species, epistatic interactions have large impacts (greater than 0.5) on four of eight traits.

## Epistatic effects

Estimating the role of epistatic interactions within the genetic architectures underlying trait divergence can help to understand how accessible a trait is to selection (Roff & Emerson, 2006). For instance, traits more closely aligned with fitness typically show more epistasis than traits less aligned with fitness (Burch et al., 2024; Crnokrak & Roff, 1995; Roff & Emerson, 2006).

Traits closely associated with fitness (i.e., reproductive traits) are usually the target of selective breeding. We often see more epistatic genetic effects and fewer additive genetic effects within the genetic architectures of these traits. Through selection on these traits, additive genetic effects are exhausted as beneficial alleles are fixed, leaving behind a genetic architecture composed primarily of epistatic genetic effects (Burch et al., 2024; Walsh & Lynch, 2018). By determining the contribution of epistatic effects to the trait divergence of these phenotypes, we were able to estimate how accessible these traits could be to selection given the absence of additive genetic effects. In Table 3, we show the observed traits in the order of their epistatic interactions, with higher numbers indicating more epistatic contribution to trait divergence. Four traits (leaf perimeter-area ratio, leaf perimeter, leaf width, and leaf area) had an epistatic contribution greater than 0.5, meaning that their genetic architectures were dominated by epistatic effects. One trait (seed mass) had 7% epistatic contribution, while the remaining traits had zero.

## Maternal effects

In our inference of genetic architectures, we show significant maternal effects for two of eight traits: leaf width and seed mass (Table 3). Leaf width shows additive maternal genetic effects dominating 37% of the trait's genetic architecture. Dominance maternal genetic effects account for 15% of the seed mass genetic architecture. Maternal effects are often inferred in analyses of traits in animals (*McAdam et al., 2002*; *Willham, 1972*). While *Cockerham (1963)* suggested that maternal effects in plants were minimal and did not require consideration, we have shown that two of the eight observed traits display maternal genetic effects as a significant proportion of their genetic architecture and that quantifying maternal effects is key to understanding the dynamics of trait evolution.

Ours is not the first study to identify significant maternal genetic effects in plants. In *Lolium multiflorum*, maternal effects were found to underlie the variation in third leaf size, as leaf length was due to additive maternal effects, and leaf width was due to interactions between maternal effects (*Edwards & Emara, 1970*). *Lolium* species show maternal effects in the size of leaves early in development, but maternal effects diminish during later stages of development (*Roach & Wulff, 1987*). A study in *Cannabis sativa* showed that maternal additive effects influence cannabichromene expression (*Campbell, Dufresne & Sabatinos, 2020*). Taken together with previous work, our study demonstrates that maternal effects are a key component of the genetic architecture of traits in plants. By quantifying the genetic architecture of traits in model plant species, we are able to improve our understanding of how accessible a trait is to selection, thereby enhancing breeding efforts across the field of agriculture.

## Compound traits

Another challenge to our understanding of genetic architecture involves our definition of traits and how it impacts inference. For instance, in this study, we analyzed eight datasets, six of which (area, perimeter, length, width, areal ratio, seed mass) can be reasonably thought of as elemental traits. We also include two compound traits (perimeter-area ratio, width-length ratio) that are measured as the quotient of two elemental traits. However, the expectation for the genetic architectures of compound traits (*i.e.*, a trait that is a function of more than one elemental trait) has not, to our knowledge, been explored. For instance, what genetic architectures will be inferred if two traits that contain only additive variation are measured and converted to a ratio and then analyzed in an LCA framework? Our study provides a single data point towards understanding what we might expect. Specifically, we found that divergence in the elemental trait of leaf width was dominated by epistatic effects (63% of total composite genetic effects) and to a lesser extent exhibited maternal effects (37% of total composite genetic effects). In contrast, the elemental trait of leaf length was composed of only autosomal additive effects. Finally, when we compare this to the compound trait of leaf width-length ratio we see that it, like leaf length, is dominated by additive effects, but both autosomal additive and cytotype additive effects are observed. However, whether this concordance between compound traits and underlying elemental traits is typical is, to our knowledge, unknown. A systematic study should evaluate the

expectations for compound traits across a range of underlying elemental trait genetic architectures.

## ACKNOWLEDGEMENTS

We thank Tahmineh Esfandani and Jennifer Elbert for assistance with the *Solanum* portion of this project and the entire Blackmon Lab for feedback on an early version of this manuscript.

### Funding

This work was supported by the National Institute of General Medical Sciences at the National Institutes of Health R35GM138098. The funders had no role in study design, data collection and analysis, decision to publish, or preparation of the manuscript.

### Grant Disclosures

The following grant information was disclosed by the authors:
National Institute of General Medical Sciences at the National Institutes of Health: R35GM138098.

### Competing Interests

The authors declare that they have no competing interests.

### Author Contributions

- Jorja Burch conceived and designed the experiments, performed the experiments, analyzed the data, prepared figures and/or tables, authored or reviewed drafts of the article, and approved the final draft.
- Crystal Nava conceived and designed the experiments, performed the experiments, analyzed the data, prepared figures and/or tables, authored or reviewed drafts of the article, and approved the final draft.
- Heath Blackmon conceived and designed the experiments, authored or reviewed drafts of the article, and approved the final draft.

### Data Availability

The data and analysis scripts are available at GitHub: https://github.com/coleoguy/plantmorph.

### Supplemental Information

Supplemental information for this article can be found online at http://dx.doi.org/10.7717/peerj.17985#supplemental-information.

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
