# Peer review of "Assessing the opportunity for selection to impact morphological traits in crosses between two Solanum species"

_PeerJ, doi:10.7717/peerj.17985_

## Round 0.1 · original submission · Major Revisions

I share the concern of Reviewer 1 that the sample size (number of plants) is not reported, and that based on the numbers of leaves in the supplemental table, it appears to be quite low. In my view, the validity of the findings are in doubt unless the authors can convincingly explain how the statistical detecting higher-order genetic effects with such a small sample size is possible, especially in the absence of F1 data.

The above is the major issue that needs to be addressed before acceptance. Inclusion of the raw data (not just the summary stats) will be important to allow critical appraisal and re-analysis.

And if a comparable experimental design & sample size has been validated before, e.g. in the simulation work Blackmon and Heath (2016), that could strengthen the case for the reliability of the findings.

The work is otherwise well-motivated and the paper is largely well-written (but please see the comments of Reviewer 2 regarding the Results and Discussion).

I leave to the author's judgement how to respond to the detailed comments about tables and figures, but agree that some streamlining and clarification would improve the presentation.

One point of clarification. In Equation 1, do the authors mean to say that "n_i" is the number of samples from a given plant and that the sum is over the number of plants (for which I would avoid using "n")?

Reviewer 1 ·

Basic reporting

English usage, references, structure are addecaute.
Raw data is available.
Figures are not always relevant:

Table 1 not relevant. The Information can be described in the text.

Table 2. Mea and Med are not defined in the legend

Figure 1 not relevant as stated for Table 1

Figures 2, 3 and 4 can be merged in a single figure

Figure 5. Legend should be self-explicative. Indicate what is P2 genome.

Experimental design

The research is within the Scope of Biological Sciences.

The objective of the experiment is to give insights on the genetic architecture of leaf and seed traits, focusing in mode of gene action, specially epistasis. The major identified knowledge gap is the low knowledge on contribution of epistasis on the genetic architecture of complex traits. This is a really relevant question in genetics, geneticist are aware that epistasis is relevant but its study is usually overlooked, mostly due to the complexity of its analysis and lack of appropriate experimental approaches to include its analysis in the experiment. Authors also studied the role of maternal genetic effects. These effects are also overlooked in most plant genetic research, so the question is very relevant.


The experimental design includes to parent genotypes (P1 and P2), F2 and reciprocal backcrosses from F1 to P1 and P2. This design is relatively simple and would allow only to descriptive conclusion, not giving insights on the genetic mechanisms underlying the possible epistasis and/or maternal effects.

The number of plants analyzed for each genotype is not indicated in the manuscript. According to the raw data, two plants seemed to be analyzed for each genotype. This is an extremely low number of plants, specially for F2 and BC populations that are segregating for a large proportion of the genome. The value of just two F2 and BC plants do not represent the trait distribution of F2 and BC populations.

Validity of the findings

Due to the limitations of the experimental design, the reported values for each population can be due sampling instead of genetic effects, specially for F2 and BCs. Therefore, it is not possible to provide meaningful conclusions.
The research should include at least 10 plants of each parent and 50-100 plants of each F2 and BCs.

Reviewer 2 ·

Basic reporting

The paper is clearly written, with sufficient description of background, methods, data collection, and statistical analisys.

The manuscript provides a comprehensive background on the importance of understanding genetic architectures in plant breeding and evolutionary biology.

The figures and tables are clear and referenced in the text. Minor point (opinion): Only figure 5 and 6 (with the core results) are truly needed. The paper is short and doesn't call for 6 figures.

I did not have access to the raw data.

Experimental design

The Methods section is detailed, the justification for the study is clear. The statistical analysis, an information-theoretic approach to account for model selection uncertainty, is a strength of the study.

Potential limitations, like the lack of F1 phenotype data and the relatively small sample sizes for some traits, could be more explicitly discussed.

Validity of the findings

The conclusions are well supported by the analysis, convincingly demonstrating significant epistatic and maternal effects for several traits, providing insights into their genetic architectures. The work could be strengthened by discussing alternative explanations (e.g. by expanding on the expected effect of unmeasured environmental variation, which they suggest could mask the genetic effects).

The Results and Discussion sections, while clear, are not particularly well written. Just as an example, the first paragraphs of each section dive into details that seem a bit out of context (as opposed to giving an overview of the results and implications).

Additional comments

Overall, the work is a valuable contribution to understanding the genetic architecture of morphological traits in Solanum.

---

## Round 0.2 · accepted · Accept

I am still sympathetic with Reviewer 1's concerns about the estimation of higher order effects based on the limited number of plants per cohort. The authors alleviate my concern somewhat by pointing out the relatively small number of parameters being estimated (as opposed to a non-line-cross QTL analysis in which there would be parameters for individual/pairwise QTL effects). I am also partially reassured by the prior work (Burch et al 2024) showing exactly how detection of epistasis is compromised by reducing the number of cohorts. But the number of leaf measurements going into each mean is still remarkably low. In the end, I will let the authors stand by their own results.

The other responses to previous reviews are satisfactory.

Minor. Please proofread the table captions. Several spaces are missing and typos present (e.g. "conûdence) in the text.

Reviewer 1 ·

Basic reporting

My major concerns are in the experimental design, please, see bellow

Experimental design

The authors analyse F2, BC1 and BC2 segregating populations, but only two plants per population. They claim that even with a such low number of plants it is possible to estimate complex genetic parameters as epistasis. Fifty- one hundred plants were proposed in the previous review, this suggestion is declare as “arbitrary” by the authors. Focussing in a F2, to study two gene epistasis, the frequency of double homozygous would be 1/16, so a large sample size would be needed to ensure minimum genotype class replications, of course, higher order epistasis would require even larger sample size. Each F2, BC1 and BC2 plant has a different combination of parent genomes, additive, dominant and epistasis effect are confounded in single plant. The effects estimated in single plants will depend mostly in sampling, depending on the genomic composition of the single plants. For that reason, quantitative genetics estimators are based on variances. I am not convinced that the experimental design is adequate to estimate the studied genetic effects.

Validity of the findings

Giving the experimental design, the conclusions about genetic effects are not clear.